

# Topological thermal Hall effect for topological excitations in spin liquid: Emergent Lorentz force on the spinons

Yong Hao Gao[1] and Gang Chen[1,2*]

**1** State Key Laboratory of Surface Physics and Department of Physics,
Fudan University, Shanghai 200433, China
**2** Department of Physics and Center of Theoretical and Computational Physics,
The University of Hong Kong, Pokfulam Road, Hong Kong, China

★ gangchen.physics@gmail.com

## Abstract

We study the origin of Lorentz force on the spinons in a U(1) spin liquid. We are partly inspired by the previous observation of gauge field correlation in the pairwise spin correlation using the neutron scattering measurement by P.A. Lee an N. Nagaosa [PhysRevB 87,064423(2013)] when the Dzyaloshinskii-Moriya interaction intertwines with the lattice geometry. We extend this observation to the Lorentz force that exerts on the (neutral) spinons. The external magnetic field, that polarizes the spins, effectively generates an internal U(1) gauge flux for the spinons and twists the spinon motion through the Dzyaloshinskii-Moriya interaction. Such a mechanism for the emergent Lorentz force differs fundamentally from the induction of the internal U(1) gauge flux in the weak Mott insulating regime from the charge fluctuations. We apply this understanding to the specific case of spinon metals on the kagome lattice. Our suggestion of emergent Lorentz force generation and the resulting topological thermal Hall effect may apply broadly to other non-centrosymmetric spin liquids with Dzyaloshinskii-Moriya interaction. We discuss the relevance with the thermal Hall transport in kagome materials volborthite and kapellasite.



Quantum spin liquid (QSL) is an exotic quantum state of matter in which spins are highly entangled quantum mechanically and remain disordered down to zero temperature [1–3]. Experimental identification of QSLs is of fundamental importance for our understanding of quantum matter. Thermal transport represents one sensitive experimental probe to unveil the nature of low-energy itinerant excitations, because other degrees of freedom, such as nuclear spins and defects, do not carry nor transport heat. Any heat current in a Mott insulator must be carried by the emergent and neutral quasiparticles [4, 5]. In the QSL regime, the deconfined spinons transport heat in the same way that the physical electrons carry charge in an electrical conductor. However, a major difficulty is that other excitations, most notably phonons, may get involved in the longitudinal thermal conductivity [6–14]. The quantitative contribution of spin excitations may be difficult to be extracted from the total longitudinal thermal conductivity due to the spin-phonon interaction, which is suggested to be present in many materials,

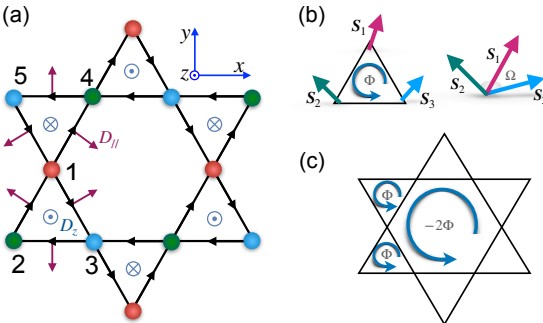

Figure 1: (a) Symmetry allowed Dzyaloshinskii-Moriya interactions between first neighbors on the kagome lattice, where $D_z$ ($D_\parallel$) is the $z$ (in-plane) component. The black arrows on the bonds specify the order of the cross product $\boldsymbol{S}_i \times \boldsymbol{S}_j$. The sublattices are labelled by colors. (b) Schematic view of scalar spin chirality for a non-collinear spin configuration, where $\Phi$ is the corresponding gauge flux through the plaquette and $\Omega$ is the solid angle subtended by the three spins. (c) Internal U(1) flux distribution induced on the kagome lattice.

especially in the ones with strong spin-orbit coupling. Thus, thermal Hall effect may be a more suitable probe to unveil the exotic excitations in QSLs since phonons do not usually contribute to thermal Hall transport.

There are *three ways* that thermal Hall effect may become signicant in a QSL. First, if the QSL is a two-dimensional chiral spin liquid, there would be chiral edge states that contribute a quantized thermal Hall response. Second, if the external magnetic field comes to modify the spinon bands such that the reconstructed spinon band develops edge states, the system would produce a quantized thermal Hall effect. A well-known example is the quantized thermal Hall effect in the Kitaev model [15] where the external field generates a Chern band for majorana spinons via high-order perturbations. This case may be not quite distinct from the first one except the first one is already a chiral spin liquid without magnetic field. The third case is when the gauge field of the QSLs is continuous. This includes, for example, spinon Fermi surface U(1) QSL [16–21], U(1) Dirac QSL [22–24], and pyrochlore ice U(1) QSL [25–28]. For the spinon Fermi surface U(1) QSL that was proposed for the weak Mott insulating organic materials $\kappa$-(ET)$_2$Cu$_2$(CN)$_3$ and EtMe$_3$Sb[Pd(dmit)$_2$]$_2$, it was suggested [18, 29] that the external magnetic field could induce an internal U(1) gauge flux through the strong charge fluctuation or the four-spin ring exchange (due to the proximity to a Mott transition) [16]. From this mechanism, the neutral spinons could experience the external field and contribute to the thermal Hall effect [30], and a fundamentally different mechanism is required to understand the thermal Hall effects in this regime. Apparently, thermal Hall effects have been observed in the kagome magnets volborthite Cu$_3$V$_2$O$_7$(OH)$_2 \cdot$ 2H$_2$O [31] and kapellasite CaCu$_3$(OH)$_6$Cl$_2 \cdot$0.6H$_2$O [32], and the pyrochlore spin ice Tb$_2$Ti$_2$O$_7$ [33]. In this Letter, we develop a theory of the topological thermal Hall effect (TTHE) for U(1) QSLs with spinon Fermi surfaces in the strong Mott regime. We will explain the emergent Lorentz force generation and TTHE for the pyrochlore U(1) QSL in a forthcoming paper [34]. In the end of this Letter, we discuss the open questions in this topic.

In the strong Mott insulating U(1) QSLs, the spinons carry emergent U(1) gauge charges and are minimally coupled to the U(1) gauge field as the spinons hop on the lattice. To twist the spinon motion, the external magnetic field has to influence the internal U(1) gauge field and then indirectly impacts on the spinon motion. In the strong Mott regime, the magnetic field couples to the spin through the usual Zeeman coupling. The internal U(1) gauge flux is related to the scalar spin chirality, $\boldsymbol{S}_i \cdot (\boldsymbol{S}_j \times \boldsymbol{S}_k)$, that involves three spins [35–37]. It is not obvious

how the linear Zeeman coupling enters to modify the three-spin scalar chirality in a disordered system, although both terms break the time reversal. A crucial observation was made by Patrick Lee and Naoto Nagaosa in the proposal [38] of detecting gauge fields or scalar spin chirality fluctuations using neutron scattering. They noticed that, with Dzyaloshinskii-Moriya interaction, the $S^z$-$S^z$ correlator contains a piece of the correlator of scalar spin chirality. Although their observation was originally made for neutron scattering, it establishes the microscopic link between the Zeeman coupling and the scalar spin chirality. In the following, we implement this observation to understand the TTHE in QSLs.

In Mott insulators where the bond centers are not inversion centers, the Dzyaloshinskii-Moriya interaction is generally allowed [39, 40]. This is a relativistic effect and is more important in the strong spin-orbit-coupled systems such as the hyperkagome material $Na_4Ir_3O_8$ [41]. A representative spin model in the strong Mott insulator has the form,

$$H = \sum_{i,j} J_{ij} \boldsymbol{S_i} \cdot \boldsymbol{S_j} + \sum_{i,j} \boldsymbol{D}_{ij} \cdot \boldsymbol{S_i} \times \boldsymbol{S_j} - \sum_i B S_i^z, \tag{1}$$

where the direction of $\boldsymbol{D}_{ij}$ is determined by the lattice symmetry from the Moriya's rule [40], and the field is applied along $z$ direction. For the kagome lattice that is used below as an example to illustrate our thought, the Dzyaloshinskii-Moriya vector for nearest neighbors can have two components [42, 43] with one normal to the kagome plane and the other in the kagome plane (see Fig. 1(a)). This Hamiltonian with variant exchange couplings on neighboring bonds has been proposed for several kagome materials where spinon Fermi surface QSLs were suggested for some materials [31, 44]. It has been estimated that the out-of-plane Dzyaloshinskii-Moriya term ($D_z$) is about 8% of the nearest-neighbor Heisenberg exchange for herbertsmithite [45]. Our purpose is not to solve for the ground state of a specific Hamiltonian. We assume that the system stabilizes a U(1) QSL with a spinon Fermi surface and explain how the spinons acquire an emergent Lorentz force from the Dzyaloshinskii-Moriya interaction.

For the spinon Fermi surface U(1) QSL, the spinon-gauge coupling is described by the following Lagrangian,

$$\mathcal{L} = \sum_i f_{i\sigma}^\dagger (\partial_\tau - i a_i^0 - \mu) f_{i\sigma} - \sum_{\langle ij \rangle} t \, e^{i a_{ij}} f_{i\sigma}^\dagger f_{j\sigma} + \int_{dr} \sum_\mu \frac{1}{g} (\epsilon_{\mu\nu\lambda} \partial_\nu a_\lambda)^2, \tag{2}$$

where the first line describes the spinon hopping on a kagome lattice and minimally coupled to the dynamical U(1) gauge field $\boldsymbol{a}$, and the second line describes the fluctuation of $\boldsymbol{a}$. The combined effect of the Dzyaloshinskii-Moriya interaction and Zeeman coupling has not been included at this stage. The connection between the emergent spinon-gauge variables and the spin variables is established from the usual Abrikovsov fermion construction with $\boldsymbol{S}_i \equiv \frac{1}{2} f_{i\alpha}^\dagger \boldsymbol{\sigma}_{\alpha\beta} f_{i\beta}$ ($\alpha, \beta = \uparrow, \downarrow$) and the Hilbert space constraint $\sum_\sigma f_{i\sigma}^\dagger f_{i\sigma} \equiv 1$. As a standard procedure, the above spin-gauge coupling can be readily obtained by introducing the gauge fluctuation to the mean-field ansatz that generates the spinon Fermi surface state [16, 17, 19]. From Elitzur's theorem, only gauge invariant variables are related to the physical spins. The scalar spin chirality is related to the emergent U(1) gauge flux $\Phi$ via (see Fig. 1(b)) [36, 38]

$$\sin \Phi = \frac{1}{2} \boldsymbol{S}_1 \cdot \boldsymbol{S}_2 \times \boldsymbol{S}_3, \tag{3}$$

where the plaquette for the flux is defined by connecting the three spins.

For this U(1) QSL, we show below that the Dzyaloshinskii-Moriya interaction and Zeeman coupling together could generate a gauge flux distribution on the kagome lattice. The Dzyaloshinskii-Moriya interaction in the spin Hamiltonian generates a finite vector spin chirality $\langle \boldsymbol{S}_i \times \boldsymbol{S}_j \rangle$. This immediately suggests the linear relationship between the scalar spin chirality

and the vector spin operator. The Zeeman coupling generates a finite spin polarization. Thus, we have a finite scalar spin chirality on the lattice. To be specific, for the kagome lattice in Fig. 1, we have

$$\langle S_2 \times S_3 \rangle = \langle S_4 \times S_5 \rangle = \lambda D_{23} = \lambda D_{45}, \tag{4}$$

where $\lambda$ is a proportionality constant with $\lambda \sim \mathcal{O}(J^{-1})$, and $J$ would be the largest exchange coupling. It is ready to see the linear relation between $S_i \cdot S_j \times S_k$ and $S_i \cdot D_{jk}$. Since we apply the magnetic field along $z$ direction, one then establishes $\langle \sin \Phi \rangle \simeq \frac{1}{2} \lambda D_z \langle S^z \rangle = \frac{1}{2} \lambda D_z \chi B$, where $\Phi$ is the flux defined on the elementary triangular plaquette of the kagome lattice and $\chi$ is the magnetic susceptibility. For the spinon Fermi surface QSL, $\chi$ is a constant. From the signs of the Dzyaloshinskii-Moriya interaction, we conclude that the induced internal U(1) fluxes by the external magnetic field on both the up triangle and the down triangle are equal and denoted as $\Phi$. The orientation of the flux loop is depicted in Fig. 1(c). Moreover, the flux through the hexagon is determined by fluxes in its six neighboring triangles. One can readily verify it equals $-2\Phi$ if adopting the anticlockwise loop convention in Fig. 1(c).

We have demonstrated that the external magnetic field induces an internal U(1) gauge flux through the combination of Zeeman coupling and Dzyaloshinskii-Moriya interaction for a strong Mott insulator QSL. This U(1) gauge flux generation differs fundamentally from the induction of the internal U(1) gauge flux from the charge fluctuations in a weak Mott insulator QSL. The induced internal flux for strong Mott insulators from our mechanism depends on the direction of the Dzyaloshinskii-Moriya interaction and is thus tied to the lattice geometry or symmetry. In contrast, for weak Mott insulators where the degrees of freedom are basically electrons, the Lorentz coupling induced flux is always uniform and does not depend on the lattice geometry. Via our mechanism, the spinon motion in strong Mott insulators will be twisted by the induced internal U(1) gauge flux. This emergent Lorentz force on the spinons generates a topological thermal Hall effect (TTHE) of the spinons. Our notion of "TTHE" is analogous to the "topological Hall effect" for itinerant magnets with non-collinear spin configurations such as skyrmion lattices that create a finite scalar spin chirality and effective U(1) gauge flux for the conduction electrons [46,47]. We recently learned the notion of TTHE was first introduced in the thermal Hall transport for magnons [48] in ordered magnets with non-trivial magnon band structure.

In the standard linear response theory to an external magnetic field, the field enters as a perturbation. For the temperature gradient, however, the Hamiltonian stays invariant while the distribution function $e^{-\beta H}$ is modified [49], thus the theoretical treatment requires some care. This difficulty is overcome by the introduction of a fictitious pseudogravitational potential as shown by Luttinger [50]. The temperature gradient is defined by $T(r) = T_0[1 - \eta(r)]$ with a constant $T_0$ and a space-dependent small parameter $\eta(r)$, that can be regarded as a space-dependent prefactor to the Hamiltonian, $e^{-H/[k_B T(r)]} \simeq e^{-(1+\eta(r))H/(k_B T_0)}$. Then, $\eta(r)H$ is regarded as a perturbation to the Hamiltonian from the temperature gradient. We can incorporate the temperature gradient into the Hamiltonian as a perturbation by using the pseudogravitional potential. Further, we assume $\eta(r)$ to be linear in the position and expand the response in terms of $\nabla \eta(r)$ since we are interested in the linear response. The energy current density can then be derived as follows, $j_\mu^E(r) = j_{0\mu}^E(r) + j_{1\mu}^E(r)$, where $j_{0\mu}^E(r)$ is independent of $\nabla \eta(r)$ and $j_{1\mu}^E(r)$ is linear in $\nabla \eta(r)$. They both contribute to the thermal transport coefficients. Ref. [49] derived the thermal Hall conductivity for a noninteracting spinless boson Hamiltonian and was often used in the literature [51,52]. Since we are dealing with fermionic spinons, so we adopt the result from Ref. [53] where a thermal Hall conductivity formula for a general noninteracting fermionic system with a nonzero chemical potential $\mu$ was obtained as

$$\kappa_{xy} = -\frac{1}{T} \int d\epsilon (\epsilon - \mu)^2 \frac{\partial f(\epsilon, \mu, T)}{\partial \epsilon} \sigma_{xy}(\epsilon). \tag{5}$$

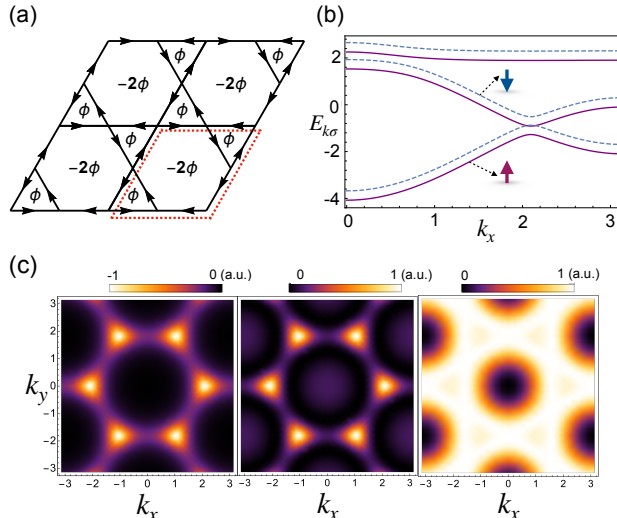

Figure 2: (a) The kagome lattice with U(1) gauge flux induced by external field through Dzyaloshinskii-Moriya interaction. The arrows on the bonds indicate the sign of the phase factor $e^{i\phi/3}$ and the flux through triangles and hexagons are $\phi$ and $-2\phi$, respectively. The area enclosed by the red dotted line is the unit cell of the kagome lattice. (b) Spinon bands for $\phi = \pi/10$ and the solid (dashed) lines are the bands for spin-↑ (-↓) spinons. (c) Density plot of the Berry curvature $\Omega_{nk\sigma}$ of the lowest, middle and highest bands for spin-↑ spinons, where we set $k_B T/t = 1$ and $\phi = \pi/3$.

Here $f(\epsilon, \mu, T) = 1/[e^{\beta(\epsilon-\mu)} + 1]$ is the Fermi-Dirac distribution function and $\sigma_{xy}(\epsilon) = -1/\hbar \sum_{\mathbf{k},\sigma,\xi_{n,\mathbf{k}}<\epsilon} \Omega_{n,\mathbf{k},\sigma}$ is the zero temperature anomalous Hall coefficient for a system with the chemical potential $\epsilon$. $\Omega_{nk\sigma}$ is the Berry curvature for the fermions and is defined as $\Omega_{nk\sigma} = -2\mathrm{Im}\langle\partial_{k_x} u_{nk\sigma}|\partial_{k_y} u_{nk\sigma}\rangle$ with eigenstate $|u_{nk\sigma}\rangle$ for band indexed by $n$ and the spin $\sigma$. Eq. (5) indicates that the thermal Hall conductivity is directly related to the Berry curvature in momentum space and a finite Berry curvature is necessarily required to generate $\kappa_{xy}$. We show below that the magnetic field induced internal U(1) gauge flux generates a finite Berry curvature and use Eq. (5) as our basis to calculate thermal Hall conductivity for the spinon metal in a U(1) QSL.

To describe the TTHE in the spinon metal, we consider a mean-field Hamiltonian for the spinon metal in the external magnetic field without including the U(1) gauge fluctuations of Eq. (2), $H_{\mathrm{MF}} = -\sum_{\langle ij\rangle}[t_{ij} f_{i\sigma}^\dagger f_{j\sigma} + h.c.] - \mu \sum_i f_{i\sigma}^\dagger f_{i\sigma} - B \sum_{i,\alpha\beta} \frac{1}{2} f_{i\alpha}^\dagger \sigma_{\alpha\beta}^z f_{i\beta}$, where the chemical potential $\mu$ is introduced to impose the Hilbert space constraint and the effect of the Dzyaloshinskii-Moriya interaction is not included here. This free-spinon mean-field Hamiltonian simply describes a QSL with a large spinon Fermi surface in the weak magnetic field. As we have explained above, the combination of the microscopic Dzyaloshinskii-Moriya interaction and Zeeman coupling induces an internal U(1) gauge flux distribution on the kagome plane. To capture this flux pattern in Fig. 1, we modify the spinon mean-field Hamiltonian by adding the U(1) gauge potential with

$$H_{\mathrm{MF}}[\phi] = -t \sum_{\langle ij\rangle}[e^{-i\phi/3} f_{i\sigma}^\dagger f_{j\sigma} + h.c.] - \mu \sum_i f_{i\sigma}^\dagger f_{i\sigma} - B \sum_{i,\alpha\beta} f_{i\alpha}^\dagger \frac{\sigma_{\alpha\beta}^z}{2} f_{i\beta}, \quad (6)$$

where we have fixed the gauge by setting the U(1) gauge field $\langle a_{ij}\rangle = \phi/3$ for all the nearest-neighbor spinon hopping in the anticlockwise manner. The net flux in each unit cell is zero (see Fig. 2(a)), so the translation symmetry of the spinons is not realized projectively.

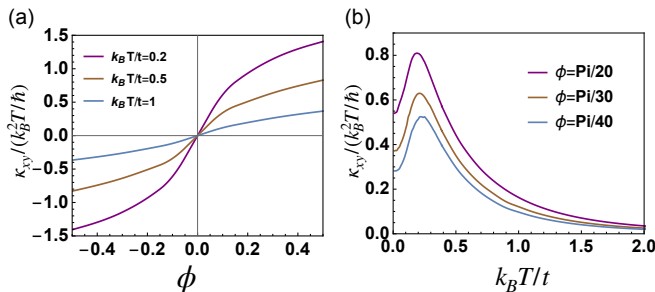

Figure 3: (a) The dependence on the induced internal flux $\phi$ of the thermal Hall conductivity at several temperatures. (b) The thermal Hall conductivity as a function of temperature.

Without the internal U(1) gauge flux, the spinon Hamiltonian $H_{\text{MF}}$ is real, and one can always choose the eigenvector $|u_{nk\sigma}\rangle$ to be real unless there is a band degeneracy, which immediately gives $\Omega_{nk\sigma} = 0$. With the internal U(1) gauge flux, the spinon Hamiltonian in Eq. (6) is complex and we expect a finite Berry curvature. Indeed as we plot in Fig. 2 for the specific choices of fluxes, the internal U(1) gauge flux reconstructs the spinon bands and creates the Berry curvatures of the spinon bands. The induced flux eliminates the band touching at $\Gamma$ point between the upper two bands and the Dirac band touching K point between the lower two bands. The Zeeman coupling further splits the spinon bands with up and down spins. Berry curvatures are enhanced at K point for the lower two bands and along the Brillouin zone boundary for the highest bands.

We calculate the thermal Hall conductivity for our TTHE based on the spinon mean-field Hamiltonian Eq. (6) using the formula Eq. (5) by varying the flux and the temperature. The results are depicted in Fig. 3. The thermal Hall conductivity $\kappa_{xy}$ vanishes at zero flux (*i.e.* at zero field) and increases monotonously with a finite flux $\phi$ in the zero flux limit. Due to the spinon Fermi surface, $\kappa_{xy}/T$ becomes a constant in the zero temperature limit [1]. The non-monotonic temperature dependence appears at finite temperatures. At very high temperatures, $\kappa_{xy}/T$ should certainly vanish because the spinons are almost equally populated and the summation of Berry curvatures of all bands vanishes, and moreover, the magnetic susceptibility would become very small at high temperatures and suppress the induced internal gauge flux. At very low temperatures, the spinon chemical potential decreases as $T$ increases. In this limit, $\kappa_{xy}/T$ can be approximated as the summation of Berry curvature of spinon bands with energies below the chemical potential [2]. As the chemical potential sits on the middle band, and the Berry curvatures of the lowest and middle bands are of opposite sign, the Berry curvature cancellation from two lowest bands becomes less, thus we would expect an increase of $\kappa_{xy}/T$ as $T$ increases. This explains the non-monotonic temperature dependence.

*Discussion*—In summary, we have proposed a physical mechanism of the emergent Lorentz force on spinons and established the resulting TTHE in QSLs. We applied this understanding to the specific cases of spinon metals in kagome lattice and calculated the TTHE. It offers a new perspective to understand the origin of thermal Hall effect of QSLs in strong Mott regime and can be related to the clear thermal Hall signal observed recently in kagome materials volborthite and kapellasite [31,32], since the main feature of the experimental $\kappa_{xy}$ in the QSL region (such as non-monotonic temperature dependence) are consistent with our theoretical result. The opposite signs of the thermal Hall conductivities in volborthite and kapellasite could arise from the opposite signs of the Dzyaloshinskii-Moriya interaction that induces the internal U(1) fluxes with opposite signs. Our theory can apply broadly to other non-centrosymmetric QSLs

---

[1] Yi Zhou, Private communications, Jan 2019

[2] See the Supplementary materials for the detailed information.

with Dzyaloshinskii-Moriya interaction and QSLs with bosonic spinons. Our understanding based on the emergent Lorentz force and/or the induced internal U(1) gauge flux through Dzyaloshinskii-Moriya interaction differs from the calculation using the bosonic spinon and Schwinger boson mean-field theory for gapped QSLs by Ref. [32] for kagome kapellasite and more recently in Ref. [55] for the square lattice. In the Supplementary Material [3], we further contrast our mechanism with the one from the strong charge fluctuation in the weak Mott regime.

Broadly speaking, thermal transport in Mott insulators is an interesting direction in quantum magnetism [30]. In the high temperature paramagnet, the high temperature series expansion can be applied. In the intermediate temperature regime where the correlation develops but there is no quasiparticle description yet, the thermal transport of these "no-particles" is an open subject in the field. The thermal transport on the pyrochlore ice material $Tb_2Ti_2O_7$ remains to be understood. In the very low temperature, various quasiparticle descriptions may emerge. For ordered magnets, magnons would be the energy carriers. The study of magnon Berry curvature has proved successful in the thermal Hall study of pyrochlore ferromagnet $Lu_2V_2O_7$ [56] and the kagome ferromagnet Cu(1,3-benzenedicarboxylate) [57]. For QSLs, the quasiparticle description is given by the parton-gauge language. Our current work about TTHE in QSL, that is based on the coupling between the spinon and the U(1) gauge field and is independent of the statistics of the spinons, elucidates the keen link between the emergent objects (such as the internal gauge flux, emergent Lorentz force and spinon Berry curvature) and the microscopic objects (such as the external magnetic field and Dzyaloshinskii-Moriya interactions) and provides the microscopic understanding of TTHE.

## Acknowledgments

This work is supported by the Ministry of Science and Technology of China with the Grant No. 2016YFA0301001, 2016YFA0300500, 2018YFGH000095 and by Grant funding from Hong Kong's Research Grants Council (GRF no.17303819).

## A  Scalar spin chirality and the instantaneous gauge flux

In this Supplementary material, following the main text, we adopt the canonical Abrikovsov fermion representation. We now consider three sites around a plaquette labeled by $i, j, k$ in an anticlockwise manner and define $P_{ijk} = \langle \chi_{ij} \chi_{jk} \chi_{ki} \rangle$. $P_{ijk}$ can also be represented by $P_{ijk} = \langle f_{i\alpha}^\dagger f_{j\alpha} f_{j\beta}^\dagger f_{k\beta} f_{k\gamma}^\dagger f_{i\gamma} \rangle$ from the definition of $\chi_{ij}$. Further defining the operator $\hat{P}_{ijk} = f_{i\alpha}^\dagger f_{j\alpha} f_{j\beta}^\dagger f_{k\beta} f_{k\gamma}^\dagger f_{i\gamma}$, simple calculations [35,37] show that

$$\hat{P}_{ijk} - \hat{P}_{ikj} = 4i \boldsymbol{S}_i \cdot \boldsymbol{S}_j \times \boldsymbol{S}_k. \tag{7}$$

The quantity $\boldsymbol{S}_1 \cdot \boldsymbol{S}_2 \times \boldsymbol{S}_3$ is the so-called scalar spin chirality and is one of the key concepts in strong correlated physics. Ignoring amplitude fluctuations, one will have

$$\langle \hat{P}_{ijk} - \hat{P}_{ikj} \rangle = \chi_0^3 \left( e^{i(\theta_{ij} + \theta_{jk} + \theta_{ki})} + c.c. \right). \tag{8}$$

The combination of $\theta_{ij}$ in the exponent is the sum of the gauge field variables around a plaquette, which is the gauge invariant flux through the plaquette. Combing Eq. (7) and Eq. (8), we can see that in the QSL state, if $\Phi$ is the instantaneous gauge flux through a triangle formed

---

[3]See the Supplementary materials for the detailed information.

by sites $1, 2$ and $3$ in a counterclockwise way, we have $\sin\Phi = \frac{1}{2}\boldsymbol{S}_1 \cdot \boldsymbol{S}_2 \times \boldsymbol{S}_3$, i.e., $\sin\Phi$ is one-half of the solid angle subtended by three spins $\boldsymbol{S}_i(i = 1, 2, 3)$, as depicted in Fig. 1 (b) in the main text. In this sense, the fluctuations in the gauge field can be interpreted as fluctuations in the chirality through each plaquette. For the kagome lattice with symmetry allowed Dzyaloshinskii-Moriya interactions (see Fig. 1 (a)), there is an important connection [38] between $\boldsymbol{S}_i \times \boldsymbol{S}_j$ and $\boldsymbol{D}_{ij}$ due to the Dzyaloshinskii-Moriya term in the spin Hamiltonian, which gives

$$
\begin{aligned}
\langle \boldsymbol{S}_2 \times \boldsymbol{S}_3 \rangle &= \lambda \boldsymbol{D}_{23}, \\
\langle \boldsymbol{S}_4 \times \boldsymbol{S}_5 \rangle &= \lambda \boldsymbol{D}_{45} = \lambda \boldsymbol{D}_{23},
\end{aligned}
\tag{9}
$$

where $\lambda$ is a constant estimated at $\lambda \sim 1/J$. It is readily verify there exists a linear coupling between the spin chirality $\boldsymbol{S}_i \cdot \boldsymbol{S}_j \times \boldsymbol{S}_k$ and $\boldsymbol{S}_i \cdot \boldsymbol{D}_{jk}$. Averaging the total chirality through the two attached up and down triangles, the in-plane component $D_\parallel$ of the Dzyaloshinskii-Moriya vectors will be canceled out and one can obtain $\langle\sin\Phi\rangle = 1/2\lambda D_z\langle S^z\rangle$. A more formal proof about the above relations can be proceeded within the first order perturbation theory [38].

## B  Wiedemann-Franz Law in the zero temperature limit

Let us now verify that the thermal Hall conductivity formula [53] Eq. (5) in the main text for the fermion systems recovers the usual Wiedemann-Franz Law for a noninteracting system in the zero temperature limit. The derivative of the Fermi-Dirac distribution function $\partial f(\epsilon, \mu, T)/\partial\epsilon$ indicates that the integral in Eq. (5) dominates around the Fermi energy. In the zero temperature limit, it represents a sharp peak and can be expanded as

$$
\frac{\partial f(\epsilon, \mu, T)}{\partial\epsilon} = -\delta(\epsilon - \mu) - \frac{(\pi k_B T)^2}{6}\frac{d^2}{d\epsilon^2}\delta(\epsilon - \mu) + \dots
\tag{10}
$$

Thus the thermal Hall conductivity is recast into

$$
\kappa_{xy} = \frac{\pi^2 k_B^2 T}{6}\int d\epsilon(\epsilon - \mu)^2\frac{d^2}{d\epsilon^2}\delta(\epsilon - \mu)\sigma_{xy}(\epsilon).
\tag{11}
$$

Using the relation $\delta''(x) = 2\delta(x)/x^2$, one can easily obtain

$$
\kappa_{xy} = \frac{\pi^2 k_B^2 T}{3}\sigma_{xy}(\mu).
\tag{12}
$$

Remarkably, $\kappa_{xy}/T = \frac{\pi^2 k_B^2}{3}\sigma_{xy}(\mu)$ suggests, in the zero temperature limit, that $\kappa_{xy}/T \neq 0$ if $\sigma_{xy}(\mu)$ is non-vanishing. In terms of the Berry curvature $\Omega_{nk\sigma}$, one can further re-express the thermal Hall conductivity as

$$
\kappa_{xy} = -\frac{\pi^2 k_B^2 T}{3\hbar}\sum_{\boldsymbol{k}, \sigma, \xi_{n,k} < \mu}\Omega_{n,\boldsymbol{k},\sigma},
\tag{13}
$$

which directly indicates that a non-zero Berry curvature is necessarily needed to contribute to the thermal Hall conductivity.

## C  Mean-field free spinon Hamiltonian

To diagonalize the Hamiltonian, by performing the Fourier transform of the mean-field Hamiltonian Eq. (6) in the main text into momentum space, one can obtain $H = \sum_{\boldsymbol{k},\alpha}\psi_{\boldsymbol{k},\alpha}^\dagger h_{\boldsymbol{k},\alpha}\psi_{\boldsymbol{k},\alpha}$,

with basis $\psi_{k,\alpha} = (f^u_{k,\alpha}, f^v_{k,\alpha}, f^w_{k,\alpha})^T$ and the matrix

$$h_{k,\alpha} = (-\mu - \alpha\frac{B}{2})I_3 - 2t \begin{pmatrix} 0 & \cos k_1 e^{-i\phi/3} & \cos k_3 e^{i\phi/3} \\ \cos k_1 e^{i\phi/3} & 0 & \cos k_2 e^{-i\phi/3} \\ \cos k_3 e^{-i\phi/3} & \cos k_2 e^{i\phi/3} & 0 \end{pmatrix}, \qquad (14)$$

where $u, v$ and $w$ are three sublattice indexes and $k_i = \mathbf{k} \cdot \mathbf{a}_i$, with $\mathbf{a_1} = (-1/2, -\sqrt{3}/2)$, $\mathbf{a_2} = (1, 0)$ and $\mathbf{a_3} = (-1/2, \sqrt{3}/2)$ the nearest-neighbor vectors. $I_3$ is a 3×3 identy matrix and the lattice constant has set to be unit. The chemical potential $\mu$ is self-consistently calculated by the following equation

$$\frac{1}{3N} \sum_{n,\mathbf{k},\alpha} n_F(\frac{E_{n\mathbf{k}\alpha}}{k_B T}) = 1, \qquad (15)$$

which guarantees the proper Hilbert space and $E_{n\mathbf{k}\alpha}$ is the dispersion of the $n$-th band of spin-$\alpha$ spinons.

## D Comparison with the weak Mott regime

For the U(1) QSL with the spinon Fermi surface in the weak Mott insulator, it was suggested that the external magnetic field could induce an uniform internal flux distribution through the strong charge fluctuation [18]. More precisely, it is through the 4-site ring exchange inter-action. As the system that was studied is a triangular lattice, the induced flux is an uniform flux on each triangular plaquette. This induced U(1) gauge flux would twist the motion of the spinons and generate a thermal Hall effect for the spinons [30]. Because of the uniform flux distribution, the translation symmetry of the spinons is realized projectively. Moreover, in principle, one would expect a quantum oscillation in the thermal Hall conductivity from the uniform flux induced by the external field [18].

In contrast, for our mechanism, the combination of the Dzyaloshinskii-Moriya interaction adn the Zeeman coupling generates a staggered flux distribution. The net flux in a unit cell of the kagome lattice is zero such that the translation symmetry of the spinon is realized unpro-jectively. Thus, one would not expect the quantum oscillation phenomena for our mechanism.

Finally, as we have mentioned in the main text, our mechanism can be well extended to other U(1) QSLs with Dzyaloshinskii-Moriya interactions. This includes both strong Mott regime and weak Mott regime. For the ones in the weak Mott regime, our mechanism and the previous mechanism from strong charge fluctuation would come together. In the strong Mott regime, however, the charge fluctuation is suppressed, and the previous mechanism can be neglected.

The QSLs in weak Mott regime that were proposed have a triangular lattice structure [16, 18]. If one can approximate the candidate organic materials as perfect or nearly perfect trian-gular lattice, the Dzyaloshinskii-Moriya interaction is absent, so one does not need to invoke our mechanism for these systems.

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
