# Peer review of "Topological thermal Hall effect for topological excitations in spin liquid: Emergent Lorentz force on the spinons"

_SciPost Physics, doi:SciPost Phys. Core 2, 004 (2020)_

## Round 1 · Referee Report · Anonymous (Referee 1) · 2019-8-1

Strengths

  1. Very readable paper, nicely written on the whole.
  2. Makes a contribution to the literature.
  3. No technical faults

Weaknesses

  1. Major weakness is whether the work is of high enough significance to warrant publication in scipost

  2. Use of the word "Topological" for this thermal hall effect (despite other papers doing similarthings) seems deceptive and should be avoided.

  3. Citation should be added into abstract.

Report

Overall I liked this paper -- it is clear and well written. My concern, however, is that it is not a sufficient advance to warrant publication in SciPost (which requires a somewhat higher level of work than, for example, PRB, in my understanding). The problem is that the results seems hardly surprising. (a) it is hardly surprising (based on symmetry) that once you add a zeeman field you will generate a chiral order parameter (b) once you have a nonzero chiral order parameter it is hardly surprising that you will have a thermal hall effect roughly proportional linear in the order parameter at least for small values. If either (a) or (b) is surprising, it is not clear to me why, and it is not explained in the manuscript. If the authors can explain this convincingly I would be happy to recommend it for publication. Otherwise, I'm not sure it is interesting enough. So although I will label the paper as "minor revision", unless the authors convince me it is interesting enough, I'm hesitant to recommend publication.

I have two further comments. First, I don't think using the word "topological" is really acceptable here. What is topological in this calculation? You use berry curvature, but this is not topological -- it is a geometric curvature, and unless you integrate over the whole zone to get a chern band, which you don't, it is not topological. There are no topological invariants, and no topological objects. The word topological is simply out of place and it seems like people just insert the word "topological" randomly in order to attract attention. I know the word has been used similarly in other publications, but this does not make it correct.

My final point is very minor. The abstract states that this work is based on prior work (second and third sentences of the abstract). These works should be cited within the abstract to clarify. With abstract citation, the entire publication information should be within the abstract itself. The way it stands it is quite hard for the reader to figure out what reference the authors mean until half-way through the paper.

Requested changes

  1. Clarify why this is important/interesting enough to be in SciPost

  2. Remove use of word "topological" where it is not appropriate (including the title)

  3. Include appropriate citation in abstract.

  • validity: high
  • significance: ok
  • originality: ok
  • clarity: high
  • formatting: excellent
  • grammar: good

Author:  Gang Chen  on 2019-08-06  [id 578]

(in reply to Report 1 on 2019-08-01)

Referee: Overall I liked this paper -- it is clear and well written. My concern, however, is that it is not a sufficient advance to warrant publication in SciPost (which requires a somewhat higher level of work than, for example, PRB, in my understanding). The problem is that the results seems hardly surprising. (a) it is hardly surprising (based on symmetry) that once you add a zeeman field you will generate a chiral order parameter (b) once you have a nonzero chiral order parameter it is hardly surprising that you will have a thermal hall effect roughly proportional linear in the order parameter at least for small values. If either (a) or (b) is surprising, it is not clear to me why, and it is not explained in the manuscript. If the authors can explain this convincingly I would be happy to recommend it for publication. Otherwise, I'm not sure it is interesting enough. So although I will label the paper as "minor revision", unless the authors convince me it is interesting enough, I'm hesitant to recommend publication.

Response: Thanks for the comment.

Let me address the referee's comment in a bit inspiring way to motivate our work. In magnetic Mott insulator, when inversion symmetry is absent, we will have Dzyaloshinskii-Moriya interaction because this is allowed by symmetry. But the physical mechanism due to the spin-orbit coupling and high order perturbation calculation from Hubbard model by Moriya in 1960 are important and decisive for our understanding of the Dzyaloshinskii-Moriya interaction in transition metal oxides.

More recently, there is a fashion of honeycomb Kitaev material. Based on the referee's logic, it is completely obvious that, any spin-orbit-coupled Mott insulator with a honeycomb geometry will have a Kitaev interaction. This is because of the symmetry. However, the contribution of Jackeli and Khaliullin's microscopic theory is important in deriving the Kitaev interaction.

The above examples are illustrative works that explain why we need a microscopic origin and microscopic understanding in our paper.

Moreover, we disagree with the referee about the statement of our work quality, we are sure that our work is above PRB.

Referee: I have two further comments. First, I don't think using the word "topological" is really acceptable here. What is topological in this calculation? You use berry curvature, but this is not topological -- it is a geometric curvature, and unless you integrate over the whole zone to get a chern band, which you don't, it is not topological. There are no topological invariants, and no topological objects. The word topological is simply out of place and it seems like people just insert the word "topological" randomly in order to attract attention. I know the word has been used similarly in other publications, but this does not make it correct.

Answer: Thanks for the comments. We used the word "topological" is based on the early works of "topological Hall effect in metals skyrmion textures". Topological Hall effect was observed and understood from the scalar spin chirality in metals with local moments. This Hall current effect is termed as "topological Hall effect".

In our context, the origin of thermal Hall effect is also from the spin texture and scalar spin chirality. The difference is that, our systems are Mott insulators, and there is no electric current but thermal current of spinons. Since our origin is from the spin chirality, so we feel it is natural to generalize this concept to thermal Hall effect. This reason that we adopt this notion was explained in page 3.

Referee: My final point is very minor. The abstract states that this work is based on prior work (second and third sentences of the abstract). These works should be cited within the abstract to clarify. With abstract citation, the entire publication information should be within the abstract itself. The way it stands it is quite hard for the reader to figure out what reference the authors mean until half-way through the paper.

Answer: Thanks very much. We are happy to do that. All works are based on the prior work.

---

## Round 1 · Referee Report · Anonymous (Referee 2) · 2019-11-10

Strengths

Paper is well written
Introduces a novel mechanism to get linear in T thermal Hall effect

Weaknesses

The mean field spinon tight binding model is derived somewhat hand-wavily.

Report

This paper introduces an interesting idea for how to get a thermal Hall effect that is linear in T at small T in the spinon Fermi surface spin liquids with U(1) gauge fields. The idea follows naturally from a few previous works, that are acknowledged in the paper, but I still find it important to highlight their mechanism as an interesting way to get substantial thermal Hall effect that scales linearly with T even in the strongly insulating regime.

My main concern is not the quality and novelty of their essential ideas, but rather with the specifics of the example that they chose to illustrate such general ideas. They consider a Hamiltonian given by Eq(1). I suspect that they are correct to assume that such Hamiltonian has a stable Spinon Fermi surface state at least at the level of Slave Fermion Mean Field neglecting the gauge field fluctuations. The pattern of fluxes that they conjecture in the presence of the Zeeman field does not increase the unit cell, so it should be easy to derive this pattern of fluxes as the one that minimises the mean field energy because their Hamiltonian is a spin bilinear. The work would be more complete if they did so and demonstrate explicitly that the pattern of fluxes is the one that minizes the mean field energy. But more than completeness, there are certain issues of consistency with their stated mean field Hamiltonian. For example, their tight binding model, Eq.(6), has conservation of total Sz. This is strange because their original Hamiltonian has DM interactions which break conservation of Sz in general. Are they assuming a restricted form of DM to get such conservation?

Their description of the Berry curvature is also rather limited. It seems they are dealing with a band structure with 3 different bands, and the bands have no touchings at finite B. What are their separate Chern numbers of each band? This is particularly important because the fully occupied bands could be contributing to the Hall effect.

When plotting Figure 3 they don't specify the value of B/t (Zeeman over kinetic energy), but only the value of the flux. These are independent parameters. What is this value?

Also I find very peculiar that their Thermal Hall conductivity reaches a value of order 1 at a temperature scale of the order of the band-width, even for very weak values of flux in their Fig. 3. For temperatures above the Zeeman scale their effect should be washed out by temperature and the system should resemble the case of no Zeeman, which should have zero Hall conductivity, because the Zeeman scale is what self-consistently determines the flux. The ban-width could be huge in comparison to Zeeman since it ie determined by J. But they never specify the value of the Zeeman scale for this plot so it is hard to tell. This is could also be an artefact of their not solving the mean field self-consistently since the value of the effective magnetic field itself should decrease with temperature because the system should have an Sz spin susceptibility that itself decreases with temperature.

Requested changes

I hope that the authors address my concerns in the Report before recommending publication.

  • validity: good
  • significance: good
  • originality: high
  • clarity: good
  • formatting: good
  • grammar: good

Author:  Gang Chen  on 2020-02-16  [id 737]

(in reply to Report 2 on 2019-11-10)

We thank the referee for his/her consideration of our paper and the detailed comments.

Referee: My main concern is not the quality and novelty of their essential ideas, but rather with the specifics of the example that they chose to illustrate such general ideas. They consider a Hamiltonian given by Eq. (1). I suspect that they are correct to assume that such Hamiltonian has a stable Spinon Fermi surface state at least at the level of Slave Fermion Mean Field neglecting the gauge field fluctuations. The pattern of fluxes that they conjecture in the presence of the Zeeman field does not increase the unit cell, so it should be easy to derive this pattern of fluxes as the one that minimizes the mean field energy because their Hamiltonian is a spin bilinear. The work would be more complete if they did so and demonstrate explicitly that the pattern of fluxes is the one that minimizes the mean field energy.

Reply: We appreciate the referee for the good advices. Actually, the usual self-consistent solution of the quadratic problem by comparing the mean-field energy has many issues, including the relaxed local constraint of fermion number on each site, particular choice of the hopping terms, higher degrees of symmetry (also explained in the following reply) and so on, thus a simple mean-field way minimizing the energy seems more or less meaningless, especially since we did not specific a particular microscopic spin model. The spinon Fermi surface state was proposed for a couple Kagome materials, and we take a mean-field model as a phenomenological description of this proposed state.

The flux pattern is not conjectured. It naturally arises from the DM interaction and the induced magnetization of the field. DM interaction does not enlarge the unit cell, thus the flux pattern preserves the unit cell. We take the flux as free tuning parameters associated with Zeeman field, which can present the existence of the spinon thermal Hall effect without violation of physics. Similar strategies are often adopted in the literatures, such as Katsura et al.’s pioneering work on thermal Hall effect in magnets.

Referee: But more than completeness, there are certain issues of consistency with their stated mean field Hamiltonian. For example, their tight binding model, Eq.(6), has conservation of total Sz. This is strange because their original Hamiltonian has DM interactions which break conservation of Sz in general. Are they assuming a restricted form of DM to get such conservation?

Reply: As it is well-known, the possible mean-field spin liquids on a lattice should be given by a systematic projective symmetry group (PSG) classification, although PSG certainly does not give complete list of spin liquid states. For the example of Kagome lattice, the DMI would certainly reduce the spin rotation symmetry from SU(2) to U(1) or Z2 at the level of the physical spin model, and it seems that we have to include some triplet terms in the ansatz. However, at the mean-field level of spinon Hamiltonian, even the spin-orbit-coupled PSG classification that involves the SOC completely breaking the SU(2) spin rotation symmetry can give a kind of mean-field spinon ansatz with full SU(2) symmetry, which is explicitly shown in Ref. [PhysRevB.96.054445 (2017)]. Therefore, at least at the mean-field level, the spinon tight binding model with conservation of Sz does not necessarily mean it is not compatible with DMI, and we can safely focus on the internal gauge flux induced by the combined effect of B and DMI. Even we consider some spin-flip spinon hopping terms due to DMI, they should have relatively small coefficients by considering the value of D/J, which only gives some quantitatively correction to the spinon dispersion, and does not influence the main result.

Referee: Their description of the Berry curvature is also rather limited. It seems they are dealing with a band structure with 3 different bands, and the bands have no touchings at finite B. What are their separate Chern numbers of each band? This is particularly important because the fully occupied bands could be contributing to the Hall effect.

Reply: In the calculation we dealt with 6 spinon bands (including spin-up and spin-down) while only showed Berry curvatures for the spin-up spinons since the Berry curvatures for spin-down spinons are a bit similar. The Chern numbers of the band can be easily calculated as -1,0,1. However, we are dealing with a gapless case, and the Fermi surface sits on the middle band, which is not fully occupied at zero temperature, thus it is not necessary to show the Chern numbers, the Berry curvatures of the occupied parts already can result in the Hall effect. In the case of the Chiral spin liquid that can exhibit a quantized thermal Hall effect, the calculation of Chern number should be necessary.

We could add some discussion of this if the referee insists on this point.

Referee: When plotting Figure 3 they don't specify the value of B/t (Zeeman over kinetic energy), but only the value of the flux. These are independent parameters. What is this value?

Reply: As we showed in the context, sin(\phi) is approximately proportion to B, they are not independent parameters. In the calculation when we vary \phi, B and thus the Zeeman splitting of bands is varied correspondingly.

Referee: Also I find very peculiar that their Thermal Hall conductivity reaches a value of order 1 at a temperature scale of the order of the band-width, even for very weak values of flux in their Fig. 3. For temperatures above the Zeeman scale their effect should be washed out by temperature and the system should resemble the case of no Zeeman, which should have zero Hall conductivity, because the Zeeman scale is what self-consistently determines the flux. The bandwidth could be huge in comparison to Zeeman since it is determined by J. But they never specify the value of the Zeeman scale for this plot so it is hard to tell. This is could also be an artefact of their not solving the mean field self-consistently since the value of the effective magnetic field itself should decrease with temperature because the system should have an Sz spin susceptibility that itself decreases with temperature.

Reply: In fact, the flux we adopted in Fig.3 (b) is not very weak, thus the thermal Hall conductivity reaches a value of order 1, and we can see from Fig.3 (a), the conductivity is certainly vanishing when the value of flux is zero. In the real materials, the actual induced gauge flux could be a bit small (for example, if DM is small) but still has the observable effect, which may give a relatively small thermal Hall conductivity.

To compare with the real materials, we need to combine different experiments to determine the effective J, DMI, effective hopping coefficient t, Zeeman scale and so on, and then carry out a complete calculation for the corresponding thermal Hall conductivity to compare with the experiment or predict the experiment. In this work, what we mainly plan to convey is a mechanism of the spinon thermal Hall effect and the influence of emergent gauge structure by Zeeman field in the strong Mott Insulators, the associated calculations in the text are to illustrate our idea, but we think the existence of the spinon thermal Hall effect is robust. We were not planning to be quantitative nor compare with concrete materials in the current practice.

List of changes:

Because Scipost requires update via arXiv submission, we were told that frequent update on arXiv does not help with the reputation. Thus, if the referee is not happy with our current response and insists on some change, we would certainly be happy to add/modify some parts and description after we receive his/her response.

Anonymous on 2020-03-11  [id 762]

(in reply to Gang Chen on 2020-02-16 [id 737])

I am satisfied with the consideration the authors have given to my earlier comments, and therefore, recommend the paper for publication.

---

## Round 2 · Author Response

Dear editor,

Since the Editor-in-charge of our recent Submission to SciPost, "Topological thermal Hall effect for topological excitations in spin" liquid: Emergent Lorentz force on the spinons" has formulated an Editorial Recommendation, asking for a minor revision. We have followed this suggestion and revised our manuscript accordingly.

Sincerely,
Gang Chen

---

## Round 2 · List of Changes

1. We have explicitly cited Ref 35 in the abstract of the manuscript. This is to follow the suggestion of the referee.

2. We have added a section in the appendix to compare the difference between the weak Mott insulating spin liquid and our mechanism for the strong Mott insulator.

3. On Page 3 of the main text, we have added the explanation of the choice of "topological thermal hall effect" in this work, and also added the citation to other references.

4. On Page 5 of the main text, we have added the referring to the supplementary materials about the comparison with weak Mott insulating spin liquids.

---

## Editorial Decision

published